# Bioplastic as a Substitute for Plastic in Construction Industry

**Ilaria Oberti * and Alessia Paciello**

Dipartimento di Architettura, Ingegneria delle Costruzioni e Ambiente Costruito, Politecnico di Milano, 20133 Milan, Italy; alessia.paciello@mail.polimi.it
* Correspondence: ilaria.oberti@polimi.it; Tel.: +39-0223995147

**Definition:** Bioplastics have proven to be a viable substitute for plastics in some sectors, although their use in construction is still limited. The construction sector currently uses 23% of the world's plastic production, both for the materials themselves and for their packaging and protection. A considerable part is not recycled and is dispersed into the environment or ends up in landfills. In response to the environmental problems caused by oil-based plastic pollution, the development of biocomposite materials such as bioplastics represents a paradigm shift. This entry aims to explain what bioplastics are, providing a classification and the description of the different properties and applications. It also lays out the most interesting uses of these materials in the construction field.

**Keywords:** bio-based material; construction; environment; circular economy; life cycle

## 1. Introduction

The problem of plastic pollution has become one of the most pressing global environmental issues. It is estimated that there are more than 269 thousand tons of plastic in the oceans, and if urgent prevention measures are not taken, there will be more plastic than fish by 2050 in terms of weight [1]. The immoderate production of plastics is growing year by year and is expected to grow by another 70% in the next 30 years [2]. The accumulation of plastics and microplastics in the oceans and on land over the decades is due to the reckless production and use of this material, which is particularly valued for its cheapness and variety of use. The causes of the plastic pollution problem can be summarized as follows: inordinate production, short use, polymer deterioration, inadequate waste management and disposal, and insufficient production chain for alternative materials. This is a great threat to the environment because already, millions of tons are dispersed in nature and end up amassing along shorelines or creating real garbage islands. This accumulation of products is also aggravated by the non-management or inadequate management of waste; in fact, it is often scattered in nature or abandoned in illegal dumps, polluting soil, fresh water, and oceans. Accumulation is also caused by the excessive use of single-use plastics: these are mainly polyethylene, found mostly in packaging and disposable products such as cutlery, glasses, or food containers. All these items have a very short life span, but a very long decomposition time that allows them to remain in the environment for years. For instance, a plastic bag has an average use of 12 min but takes 500 years to fully decompose [3]. Plastic pollution is aggravated by the deterioration of polymers, which produce a high amount of plastics/microplastics in the environment. Contributing to the waste management problem is the fact that only a small percentage of plastic is recycled. This is due to both the high economic cost of recycling processes and the low financial incentives for recycling. It is much cheaper to produce most types of plastic from scratch than to recycle old plastic. In addition, the complexity of recycling processes is an obstacle to achieving circular economy goals. Finally, the scarcity of production chains of more sustainable materials must be considered. Ecological alternatives are still limited and mainly related to products such as packaging, containers, and single-use plastic items, for which the raw materials used are plant-based polymers or polymers made from animal proteins. In most cases, the extraordinary properties of plastics mean that other, more environmentally friendly

options are in short supply. However, a shift to more sustainable, non-petroleum-derived materials is the key step that production chains should take to reverse the trend of increasing plastic use and to transform the current linear economic model into a circular one. In fact, the circular economy is based on the different cycles of the production chain of discarded products to limit waste, minimize the use of totally new raw materials, maximize the potential of a material, and reduce the amount of waste that is dispersed into the environment. In this economic and production model, bioplastics assume high importance, particularly all those that are compostable since, after being used in different production cycles, they can be biodegraded, avoiding remaining in nature for years as opposed to fossil plastics. With this in mind, biopolymers, and biomaterials more generally, should be considered as a means for the ecological transformation of the economy and the construction sector. However, as Figure 1 shows, only 8% of the bioplastics produced are used in the buildings and construction sector, compared to 23% of the use, in the same sector, of plastic produced worldwide [4].

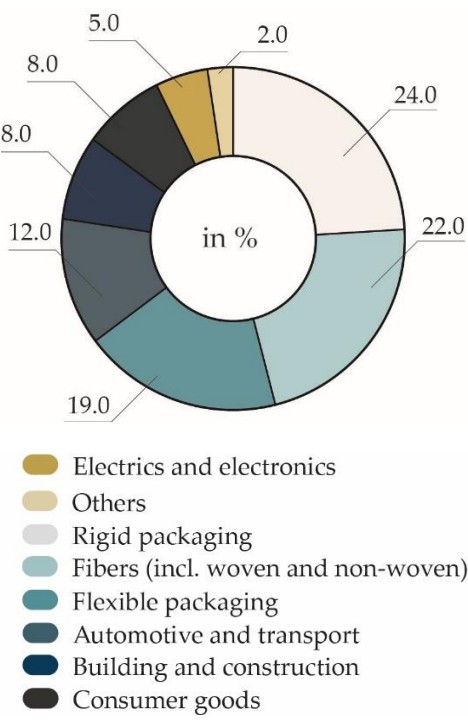

**Figure 1.** Use of bioplastics by market segment (2021). Reworked by the authors from [5].

The inordinate consumption of non-renewable raw materials and the production of waste that is difficult to degrade are the main problems associated with the use of plastics. Adherence to a production model using bioplastics would counteract both of these problems by switching from the exploitation of non-renewable raw materials derived from oil to renewable raw materials often derived from the agricultural sector. However, reference to the two aspects, raw materials and waste, is not sufficient to determine whether bioplastics are really a solution to the problem of plastic pollution.

## 2. Bioplastics

### 2.1. Bioplastic Definition

To understand the concept of bioplastics, it must first be made clear that this, like "traditional" plastics, includes materials composed of polymers. Polymers can be divided into two categories according to the origin: They are either of natural origin when they generate naturally and are, therefore, derived from biological processes, such as cellulose or chitin; or of synthetic origin, when they are produced artificially by humans through chemical processes. From this, it can be seen that not all plastics composed of synthetic polymers are part of the category of materials commonly referred to as "plastics", but

rather fall under the category of bioplastics, since, being biodegradable, they meet the second characteristic of the official definition given by European Bioplastics. The terms "polymer" and "biopolymer" are commonly misused, using the first as a synonym for "plastic" and the second (with the intention of indicating a natural polymer) as a synonym for bioplastic. In general, bioplastics offer additional benefits over plastics obtained from oil, such as reduced carbon footprint, better functionality, and provide additional options in waste management, such as organic recycling. The use of bio-based plastics can reduce the consumption of non-renewable raw materials, the price of which may increase significantly in the coming decades due to increasingly limited availability. In addition, these plastics have the advantage of reducing greenhouse gas emissions because the plants that are grown for their production contribute to the absorption of $CO_2$ from the atmosphere during their growth period. Certainly, all of the aspects mentioned above are positive elements, but in order to be able to fully assess the situation, other aspects must also be taken into consideration. In fact, if the sector of bio-based plastics were to become widespread, one would have to fight from the outset against the exploitation of land used for the intensive cultivation of cereals [6] such as corn, to the detriment of agricultural food production and with the strong risk of compromising natural habitats. The Drivers of Deforestation and Degradation study [7] shows that among the main causes of global deforestation is agriculture, especially intensive agriculture for industrial purposes, which causes the alteration of local ecosystems, with possible consequences for fauna and flora. In addition to environmental concerns, the use of bio-based plastics raises some doubts regarding the durability, rigidity, and strength of these materials. It is possible, in fact, that these tend to lose their functional properties under certain environmental conditions such as high humidity. Therefore, there is still a need to improve the performance of bio-based materials to make them suitable for applications in the construction field.

*2.2. Classification*

The term bioplastic refers to a set of materials with different properties and applications, which can be grouped into three macro categories. The official classification follows the definition given by European Bioplastics, which distinguishes between bio-based polymers, biodegradable, or both [8]. An important clarification to make is that the adjective "bio-based" is not synonymous with the adjective "biodegradable", with the latter indicating a material that is degraded as a result of the action of certain microorganisms. As shown by Figure 2, some bio-based materials are biodegradable, for example, polylactic acid (PLA), while others are not, such as bio-PET.

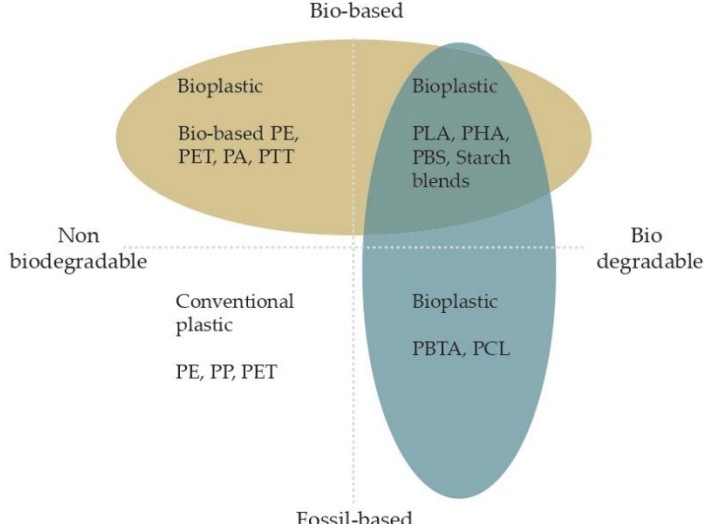

**Figure 2.** Classification of bioplastics according to European Bioplastics [9].

Bioplastics, therefore, are:

- Bio-based, derived from renewable natural raw materials;
- Biodegradable, of synthetic origin, but degrades quickly;
- Bio-based and biodegradable, when it possesses both properties (natural origin and degrades quickly).

### 2.2.1. Bio-Based Plastics

Bio-based plastics refer to those materials or products that are wholly or partially derived from organic biomasses (bio-based materials), such as corn, sugarcane, and cellulose.

### 2.2.2. Fossil-Based but Biodegradable Plastic

All plastics that are derived from fossil raw materials but degrade rapidly belong to this category. Actually, the composition of a fossil-based plastic does not preclude the quality of biodegradability. There are plastic polymers derived from oil that, under optimal conditions, decompose more rapidly than similar organic polymers derived from biomasses. All those oxo-biodegradable plastics are excluded, i.e., conventional plastics to which a mixture of additives such as metal salts are added during production, which accelerate the degradation process in the presence of ultraviolet light, oxygen, or heat. This exclusion, proposed by European Bioplastics, is supported by the results of a study conducted by researchers at Michigan State University [10].

### 2.2.3. Bio-Based and Biodegradable Plastics

Bio-based and biodegradable plastics are, as in the first case, materials or products that are wholly or partially derived from organic biomasses, but with the special feature of being biodegradable, such as polylactic acid (PLA), polyhydroxyalkanoates (PHAs), polybutylene succinate (PBS), and starch blends.

### *2.3. The Original Raw Materials*

Fossil-based bioplastics are derived from the combination of various organic compounds produced from petroleum or hydrocarbons such as propane and butane, which are contained in natural subsurface gases or oil. In contrast, most known bioplastics are derived from carbohydrate-rich biomasses. The biomasses used can have different origins; they can come from the agricultural sector or the food industry, but there are other sources. Depending on the type of biomass used, the type of bioplastic also varies. Biomass can be divided into three macro classes [11]: those of first, second, or third generation. The first-generation feedstock includes food crops of carbohydrate-rich plants such as corn, sugarcane, beetroot, potato, tapioca, and barley. Starch and dextrose sugars are extracted from these plants. Many of the bioplastics mentioned are produced from these biomasses, as they are among the most efficient for producing bio-based plastics. The second generation of biomass includes materials with high lignocellulosic content not intended for food production. These substances represent the inedible by-products of food crops such as straw, corn stover, sugarcane waste, wood, or plant fibers, including hemp and flax. Bioplastics produced from these substances are biodegradable and compostable. Finally, further development occurs with third-generation raw materials. These are derived from the above-ground cultivation of algae, fungi, and microorganisms, which process the waste biomass. By using this family of materials, it is possible to contain the problems related to land consumption that are very pronounced in the first-generation biomass. To date, the construction industry has developed according to a logic based on oil extraction; examples include concrete, metals, and plastics. What is needed is a change of mindset that pushes toward a cultivated building materials approach. The "Cultivated Materials" concept implies the possibility and desire for a closed-loop building sector, where raw materials are derived from biomasses that can be returned to the soil after use. The idea behind this approach is that if a building can be cultivated, that is, whose materials that make up the various building components are derived from a cultivation process, it could also be composted

at the end of its life and, thus, become the source of a new cultivation process [12]. A lot of cultivated/natural materials, like wood, were used in the past, nowadays substituted by other materials with improved properties. The challenge for the future is to recover the approach to cultivated materials, adapting them to the needs of current times, taking advantage of new technologies and knowledge.

### 2.4. End-of-Life Options

The European Commission has established a hierarchy of steps and alternatives to which products can be subjected before irreversible collection for landfill, which remains the last choice. To overcome the problem of pollution due to the accumulation of waste in the environment and to alleviate the waste load that goes to landfills every day, there are several alternatives that contribute to better disposal. Efficient waste management is of paramount importance to the European Commission, which, through Directive 2008/98/EC, has defined the five stages of waste disposal, classifying the treatments to which waste can be subjected according to its characteristics [13]. The advantage of using bioplastics is that organic recycling is used as a method of waste disposal. In fact, while the other types of end-of-life of a product can also be applied to conventional plastics, organic recycling is a prerogative of biodegradable bioplastics. Based on the model described, the priority treatment in waste management is the prevention and minimization of resource use, maximizing the functional performance of the end product, such as the number of times an object can be reused without losing its performance. Precisely in this regard, the European Commission issued Directive 904/2019, known as Single-Use Plastic (SUP), banning plastic disposable products in all EU countries from January 2022. The first end-of-life scenario for products is recycling; this practice can only be implemented through separate collection of plastic and the wet fraction. Recycling can be of three types: mechanical, chemical, and organic. The latter is practicable only for the disposal of bioplastics. When bioplastics are biodegradable, three new organic recycling options open up: home composting (for unprocessed or treated organic matter), industrial composting, and anaerobic digestion. Organic recycling includes "circular" end-of-life options for biodegradable or compostable plastic products. These are materials that biodegrade over a specific period, under particular conditions of temperature, moisture, and in the presence of microorganisms. The European standard EN 134325:2002 defines the requirements a product must satisfy to be considered biodegradable: it is defined as such if it dissolves 90% within 6 months and breaks down into simple molecules such as water, $CO_2$, and methane. If a material is biodegradable, however, it is not necessarily also compostable; in fact, these two terms are not synonymous. A compostable product is not only biodegradable, but as it degrades it turns into compost, which can be used in agriculture as a natural fertilizer by decomposing at least 90% within 3 months. If the previous options are not available, the best solution turns out to be energy recovery through waste-to-energy: bioplastic waste is incinerated, generating partially renewable energy through heat recovery [14]. When bio-based plastics are involved, incineration of these releases the same amount of carbon dioxide originally absorbed by plants, thus closing the loop. Because of its inherent environmental risks, landfilling is, according to the waste hierarchy, the least viable end-of-life option. Although these materials can be effectively recycled, in many countries, landfilling is still the most common disposal solution.

## 3. Applications in Construction

### 3.1. Biopolymers as Admixtures in Concrete and Mortar

One of the main applications of bioplastics in the construction industry concerns their use in concrete mixtures and dry premixed mortars, as additives that optimize these products. In several applications, bio-based aggregates compete on par with those of synthetic origin; their market is, therefore, likely to expand, especially with the increasing advances achieved by technology. The advantage of adding additives in building materials has been known since ancient times. If the main components of the mixture cannot

provide the required properties, then it is necessary to implement these by introducing new components. Depending on the properties required, specific additives are added into the mixtures. The main applications of bio-based admixtures are concrete mixtures and mortars, although they are also used in paints, exterior and interior coatings, gypsum board, stucco, and joint mortars. Concrete is among the most commonly used building materials by humans, so it constitutes an important market for admixtures: about 15 percent of the total volume of concrete contains them [15]. Organic admixtures that are used in concrete mixtures include lignosulfonates, protein hydrolysates, and Welan rubber. Lignosulfonates constitute, by volume, the largest category of admixtures used in the construction industry. These can be used either for plasticization of concrete by improving its fluidity and workability, or water reduction; thus, high mechanical properties and durability can be obtained. In precast concrete, lignosulfonates are used to achieve greater mechanical strength. Other admixtures used include protein hydrolysates, which are used to lower the surface tension of water significantly in the preparation of expanded concrete. Protein hydrolysates generate spherical foam bubbles, which provide about 20 percent more compressive strength than hexagonal bubbles produced by synthetic foams [15]. Consequently, protein hydrolysates are preferred where the need arises to foam concrete to achieve low specific gravity while also maintaining compressive strength. A significant example is also found with the use of Welan rubber in concrete preparations. It is used as a viscosity modifier to increase the performance of highly fluid concretes, such as self-compacting concretes, which have a tendency to disintegrate, indicated by the formation of a water layer at the surface, or by the settling of large aggregates. Its main advantage is to stabilize without nullifying the fluidity of the mixture [15]. Additives are also used to enhance the performance of mortars, mainly used for plasters, tile adhesives, and self-levelling underlayments. Both masonry plasters and tile adhesives require a water-retention agent to prevent water loss into the wall: among bio additives, cellulose ethers are the most widely used for this purpose. Other uses include gypsum board, which requires a dispersant to thin the gypsum slurry. In this field, ammonium lignosulfonate is the most widely used. For grouts, on the other hand, the addition of xanthan gum allows fluidity to achieve better penetration of the product into the crack. A more recent study has furthered research in this area, identifying additional biopolymers that can be used as admixtures in the production of sustainable concrete. Tests carried out by Karandikar using potato starch showed increased mechanical strength of lime mortars containing this natural polymer [16]. Again, Govin et al. documented that guar gum increases the water retention capacity of cement mortars, limiting water loss [16]. These same properties have also been identified in a viscous biopolymer derived from Opuntia Ficus Indica extract. It has been shown that the use of this natural polymer results in a significant improvement in durability (about 5–7 times) in lime mortars [16]. The biopolymer has been used as an additive in a hydraulic lime mortar, with positive effects on mechanical properties due to the adhesive nature of this material. Among the natural polymers mentioned above, cactus extract is an interesting alternative because of its ability to both improve the mechanical strength and durability of lime mortar. However, to fully understand the potential of its use, it is necessary to analyze in more detail the interactions created between the viscous biopolymer and the cement mixture. Based on the results obtained, it is possible to state that these admixtures improve the water retention of concrete, preventing premature drying of the mix, thus reducing the possibility of cracking. Nevertheless, the possible negative effects must also be considered, especially on the setting and hardening of concrete.

### 3.2. Cement Replacement: Starch and Alginate as Adhesive Components

A study conducted by the University of Delft [17] seeks sustainable alternatives to replace cement as a binder in building materials. The subject of the research is the possibility of using certain biopolymers, particularly cornstarch and alginate, no longer just as additives, but as binders. In fact, the concrete mix is composed of a binder, cement, aggregates that vary in grain size, additives, and water. Cement, however, has a significant

impact on the environment, so the exploration of new bio-based building materials as a binder is a step toward more sustainable construction. The analysis conducted in [17] is based on experiments showing that mixing cornstarch with sand and water and heating (to a temperature not exceeding 200 °C) the mixture produces a hardened solid material, called CoRncrete, with compressive strengths with values up to 20 megapascals. During heating, the starch molecules partially dissolve and form a gel that glues the sand grains, hardening when dried. Alginate is another biopolymer that can be used as a binder. It is extracted from a wide variety of algae that inhabit the oceans. The advantage of using alginate, as opposed to cornstarch, is that the raw material from which it is extracted grows in the sea and does not require land area to grow. The quick drying and low weight of CoRncrete along with its derivation from renewable sources make it an attractive material. CoRncrete can be used as a block structural material, such as bricks, in dry environments placed in arid areas and inside buildings. Mortar to seal the joints of the blocks could be replaced using fresh CoRncrete heated in place. Alternatively, the blocks could be molded through molds into interlocking shapes. Given the low temperatures required for heating, CoRncrete is an attractive material for 3D printing buildings. Although the temperatures required to heat and dry this material are much lower than in traditional concrete production or clay brick making, the heating procedure cannot currently be applied to the scale of a building [18]. As for high degradability, this has both positive and negative effects: although it cannot be used outdoors in wetlands, "biocement" can find application in temporary structures. In fact, alginate or cornstarch biopolymers are resistant to compression as long as they are dry; however, once exposed to water, they weaken easily. Further development is needed to improve the durability of such materials so that their applications in construction can be consolidated and expanded. The use of additives or hydrophobic coatings are viable options to improve their durability, if these do not compromise degradability or recyclability factors, which are necessary to counter the waste problem. The eco-cost of CoRncrete, calculated through a cradle-to-cradle life cycle assessment (LCA), appears to be higher than traditional concrete [17]. This is mainly due to the use of fertilizers for growing corn, which results in higher ecotoxicity. If these aspects were eliminated, CoRncrete could offer potential reductions in environmental impact. Based on these results, it can be concluded that CoRncrete is a promising building material due to its light weight, good compressive strength, biodegradability, and its derivation from renewable sources. However, factors such as durability and sustainability pose challenges for its use.

### 3.3. Application of Biopolymers as Aggregates in Concrete Mixtures

Lightweight concrete is a mix containing lightweight aggregates whose density is lower than the aggregates used in traditional concrete. It is made by replacing all or part of the stone aggregates (gravel, crushed stone) with lighter aggregates that have a honeycomb structure. Such aggregates can be either artificial (naturally derived but obtained with an industrial process), such as expanded clay, pumice, and perlite, or synthetic, such as expanded polystyrene (EPS). The main disadvantage of using artificial aggregates is the exploitation of limited, non-renewable resources, resulting in the depletion of the planet. In fact, once incorporated into the concrete mix, all aggregates undergo irreversible transformation. On the other hand, the use of synthetic aggregates, such as EPS, derived from petroleum products, involves the use of non-renewable raw materials and a serious environmental impact due to the extremely polluting extraction and processing processes: the emission of pentane during the production of EPS is another fact that has considerable effects [19]. In addition, expanded polystyrene is non-biodegradable and resistant to photolysis. All these reasons, added to the rising price of crude oil and its derivatives, are causing a growing interest in the development of more environmentally friendly materials. The study conducted by the University of Auckland [19] investigated the possibility of using expanded polylactic acid (EPLA) as an aggregate for lightweight concrete to replace EPS. Polylactic acid is a biopolymer derived from cornstarch and sugarcane, expanded using carbon dioxide so as to drastically reduce its density. Experimental research shows positive

and negative aspects: EPLA is a biodegradable biopolymer that is less polluting than EPS, but when used in concrete, it reduces its compressive strength and elastic modulus. The data cannot be considered conclusive, and further studies are needed to ascertain the validity of this alternative.

*3.4. Reinforcing Biopolymers for Earthen Building Construction*

Biopolymers can be used in earthen constructions in order to improve certain properties. Earthen constructions have been a type of housing used since ancient times due to the easy availability of the raw material. Today, they still constitute housing for about one-third of the world's population [20]. However, these buildings have several limitations in terms of mechanical strength and durability. They are particularly susceptible to intense weather phenomena such as rainfall and flooding, the damage caused by which poses a potential risk to humans [21]. The use of binders or reinforcing substances is, therefore, necessary to increase the stability of earthen constructions. The main binder used by humans in construction is cement; however, the process for its production has been reported to be the source of nearly 5 percent of global greenhouse gas emissions [20]. Even when used in a limited dosage, cement can significantly increase the carbon footprint of earthen construction. In particular, a study by Van Damme and Houben [22] showed that "the ratio of carbon dioxide emissions to compressive strength is greater in cement-stabilized unfired earth than in other building materials such as concrete or unstabilized unfired earth." A further disadvantage to the use of cement is major regional differences in the global distribution of its price: it is particularly expensive in developing countries, making it even more difficult to use in those geographic areas. For these reasons, numerous studies have been conducted since the early 2010 s to investigate the stabilizing effects of biopolymers in earth-based materials to identify more sustainable alternatives. In 2012, Chang and Cho [23] studied the effect of glucan biopolymer on the compressive strength of soil. They showed that adding a small amount (less than 5 g/kg) of the biopolymer under study results in a threefold increase in compressive strength of soil, while adding 10 percent cement only leads to a doubling. Gel-type biopolymers, particularly gellan gum and xanthan gum, have also been used as stabilizers in earth-based constructions. Because of its stability, even under high temperature and low pH conditions, gellan gum is used as an additive to improve strength. Similar performance is also obtained with the use of xanthan gum. The results obtained meet the strength criteria for these two biopolymers to be used as a binder in the production of rammed earth bricks. Xanthan gum is also used in combination with natural fibers to make an even more efficient building material. Earth remains a brittle mineral compound, which possesses low tensile strength and is subject to shrinkage during drying, resulting in fractures; therefore, the addition of natural fibers such as hemp, flax, and date palm is a solution to cope with these problems. The results collected [24] show that the addition of fibers alone decreases the compressive strength of the soil compared to the sample that does not contain them. In contrast, the combined addition of xanthan gum and natural fibers creates a synergistic effect that significantly improves the properties of the soil sample, giving it higher compressive and tensile strength and better ductile properties. To increase the poor water resistance of earthen architecture, a study conducted by Perrot, et al. [25] investigated the feasibility of using chitosan as an additive or material to produce an exterior coating to improve the resistance of rammed earth and adobe blocks to water-induced degradation. One of the most attractive aspects of chitosan concerns its low cost since it is commonly extracted from crustacean waste from the food industry. In addition, it has proven properties such as biodegradability, antibacterial activity, and nontoxicity. In the aforementioned study, the soil material samples, both treated with the biopolymer and untreated, were subjected to a number of tests to evaluate their compressive strength, tensile strength, and flexural behavior, as well as sensitivity to water-induced degradation. The outcome of the tests showed a positive influence of

the presence of chitosan both as a surface coating and as an additive added to the soil mixture. Low concentrations are sufficient to make a water-repellent outer coating while, when incorporated into the manufacture of the mixture, the results show that at least 3 percent is necessary to provide some hydrophobicity. In summary, the analysis shows that even low concentrations of chitosan can greatly improve performance with regard to water-induced degradation and mechanical properties. However, it is important to continue studies concerning the effectiveness of chitosan about long-term durability.

### 3.5. Biopolymers as a Material for 3D Printing

3D printing, also known as additive manufacturing (AM), consists of a series of processes for fabricating components by adding material one layer after another, each corresponding to a cross section belonging to the 3D model. This technology enables the production of complex and irregular structures, thanks to the ability to design and customize the digital model that will later be printed. Although the most commonly used materials for 3D printing are plastics, metal, and ceramics, a number of examples demonstrate the possibility of using bioplastics as a material for 3D printing, also in the field of buildings and construction [26]. Substances such as lignocellulose, starch, algae, and chitosan-based biopolymers are possible naturally derived alternatives for the application of additive printing to the construction sector through an environmentally friendly approach. Polylactic acid is used as a bioplastic filament for 3D printing, proving suitable due to its low melting temperature. In addition to PLA, numerous other bioplastics designed and patented specifically for specific projects have been applied over the past decade. Berlin-based start-up Made of Air has developed a bioplastic composed of forest and agricultural waste that is combined with a binder derived from sugarcane to create thermoplastic granules that can be melted and molded. This material absorbs carbon and can be used for different applications from furniture to building facades. Since the material stores more $CO_2$ from the atmosphere than it emits during its life cycle, Made of Air is a carbon-negative material. This product was first installed on a building in April 2021, to clad an Audi dealership in Munich with hexagonal panels called HexChar. Earthen constructions are traditional structures for many countries around the world, despite for which development is still limited due to the long time required for labor and curing of the material [19]. For these reasons, recently, some methods have been analyzed to make the construction process easier and faster, to achieve a construction process of traditional earthen architectures compatible with modern standards in terms of productivity and cost. As a result, 3D printing was also applied in the production of earthen buildings, in an attempt to combine state-of-the-art techniques with the most common and ancient material in the history of construction. To achieve its goals, alginate biopolymer was added to the earth to increase the setting speed of the material and make its extrusion more efficient.

### 3.6. Cultivating Building Elements: A Bioplastic That Is Grown

Particularly promising for the future are those bioplastics produced from third-generation raw materials, waste biomass from food processes, which do not involve land consumption. Producing a new sustainable material from waste products makes it possible to minimize environmental impact by avoiding the burning of agricultural waste, which is a major source of pollution, especially in developing countries [27]. A number of studies, first carried out on packaging production, demonstrated the ability of organisms such as algae, fungi, and microorganisms to process waste biomass by transforming it into a bioplastic, capable of fostering a zero-waste industry. Over the years, research has focused on the potential of the fungal mycelium, the vegetative apparatus of fungi, which is able to grow thanks to the lignocellulosic content present in agricultural waste material [27]. The cellulose contained in the biomass acts as both food and structure for the growth of the mycelium which, solidifying in about a week, goes on to unite all the smaller portions of the tissue forming a larger whole. The fungal tissue acts as a binder, and as it solidifies, it will harden into whatever form chosen, constituting a light and strong material. The cellular structure of the mycelium consists of chitin polymers, a feature that distinguishes fungi from vegetative matter and gives the

material plastic behavior once processed. For this reason, in tests carried out [12], materials grown from fungal mycelium compare favorably with some plastics and other composites made from fossil-based resins and glues. There are numerous interesting features of mycelium-based bioplastic; in fact, it is 100 percent compostable and is obtained through a process that requires no energy and releases no carbon emissions. In addition, mycelium-based bioplastic can be made from different types of agricultural waste, depending on the availability of the location, making it possible to produce it in any region of the globe from local biomass remains. For example, Pelletier et al. made mycelium-based bioplastics from waste products such as grass, rice straw, sorghum, flax, and hemp, while Appels et al. used agricultural wastes such as straw, sawdust, and cotton [27]. Empirically, some properties of the new material have been observed that make it particularly promising for use in the construction sector, especially in the production of sound-absorbing, heat-insulating and bio-brick panels. Through incidental tests, in fact, Hari Dharan [12] has shown that the bioplastic analyzed has high fire resistance, a good index of both thermal and acoustic insulation, and is free of toxic volatile organic compounds. In addition, the compressive strength and flexural characteristics are comparable to those of fossil-based compounds and those of engineered wood. These findings inspired the cultivation of some bricks, composed of sawdust and mycelium (Figure 3), later assembled in Philip Ross's design for a tea house exhibited in Germany.

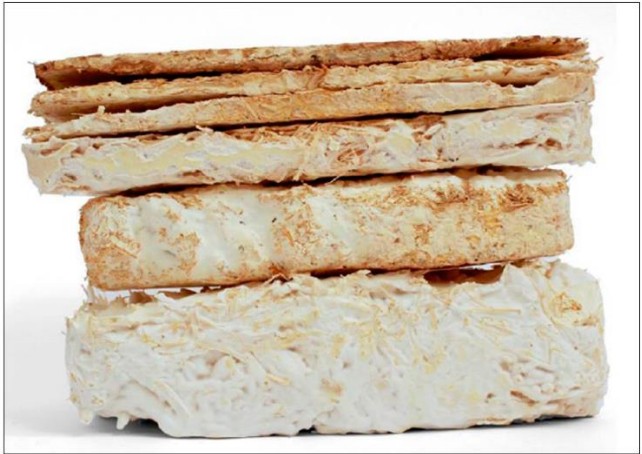

**Figure 3.** Mycelium-based composite (Source: Wikimedia Commons).

The spread of knowledge about this material led to several experiments: in the New Children's Museum in San Diego, a room composed of 1200 shaped mycelium bio-bricks was set up for an interactive exhibition. These blocks were created to be used in hands-on games, as if they were Lego bricks. When the initiative ended in 2014, the museum earmarked all the bricks for San Diego's municipal composting program, leaving no waste behind. The use of mycelium is not only about the materials themselves, but also involves a rethinking of standard production methods. The manufacture of each individual component no longer involves mass industrial production. The material is cultivated and needs its own development time to which it forces us to adapt. Mycelium-based plastics can be used for a variety of purposes: they can be grown to make a monolithic element, through the use of large-scale formwork; but they can also be made into individual building elements, later assembled mechanically as in David Benjamin's Hi-fi project. These individual elements can be used both in the structural field, through the production of bio-bricks, and for the production of panels for thermal and acoustic insulation. At the same time, research on these materials is producing knowledge that allows the materials themselves to inspire and transform architectural design, changing its aesthetics, but more importantly, the thinking behind it. Today, interdisciplinary teams are working in this emerging field to integrate fungal bioplastics into the construction industry [28]. As a prototype, the Mushroom Tiny House, a small mobile home that exceeds just 5 $m^2$ of usable area and possesses high energy performance, was made by Ecovative. The insulation,

interposed to two layers of pine wood (Figure 4), grows in a few days, generating a light and homogeneous compound, comparable to classic polystyrene [29]. Other interesting uses of mycelium in architecture can be found in projects such as Breeding Space by Maria Mello [30] or Grown Structures by Aleksi Vesaluoma in collaboration with Astudio [31].

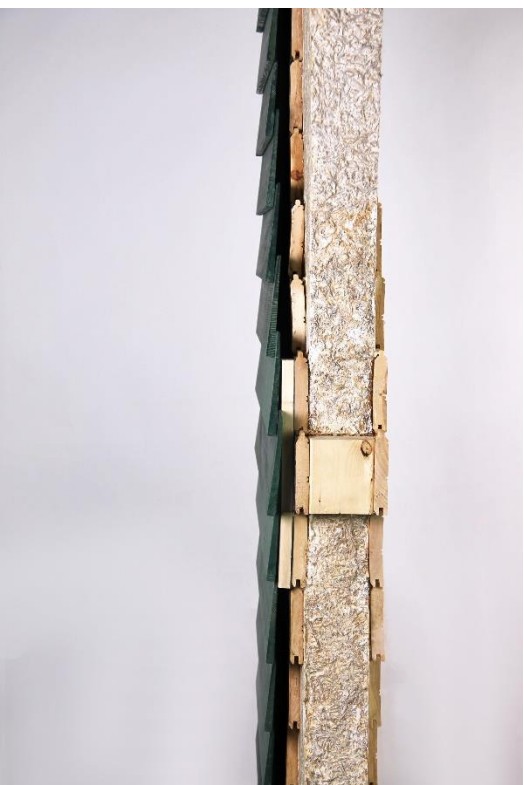

**Figure 4.** Mycelium-based composite. (Source: Courtesy of Ecovative).

## 4. Conclusions

Bioplastics provide an attractive sustainable alternative to some construction materials, contributing to solve the main problems caused by the use of plastics, such as the consumption of non-renewable raw materials and the production of non-biodegradable waste. Despite the many advantages of bioplastics, such as being bio-based, biodegradable, or both, not all aspects are positive and, therefore, their disadvantages must also be considered. However, there is a need to further study and test these new materials for application in the construction sector in order to make the most of their many potentials. To bring about a transformation, however, it is not enough to replace traditional plastics with bioplastics: a substantial change in the economic and production model is required. It is urgent to complete the transition from a linear to a circular economy model, designed to be regenerative.

**Author Contributions:** Conceptualization, I.O.; investigation, A.P.; resources, A.P.; writing—original draft preparation, A.P.; writing—review and editing, I.O. and A.P. All authors have read and agreed to the published version of the manuscript.

**Funding:** This research received no external funding.

**Institutional Review Board Statement:** Not applicable.

**Informed Consent Statement:** Not applicable.

**Data Availability Statement:** Not applicable.

**Conflicts of Interest:** The authors declare no conflict of interest.

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
