# Peer review of "Bioplastic as a Substitute for Plastic in Construction Industry"

_encyclopedia, doi:10.3390/encyclopedia2030095_

Round 1

Reviewer 1 Report

In this paper, an overview of the possible use of bioplastics in construction fields is reported. This is doubtless an interesting and contemporary topic.

Text needs some corrections:

-       In the definition it is suggested to replace the sentence: “This article aims to give a definition to bioplastics, not as a single material, but by providing a classification of the entire family of materials with different properties and applications.” with: This article aims to explain what bioplastics are, providing a classification and the description of the different properties and applications;

-       Lines 21 and 22: replace “seas” with “oceans”;

-   Sentence lines 22-25: the subject of the problem of plastic/microplastic in the oceans must be separated by the properties of polymers;

-       Line 24: replace “its long life” with “its low density”; durability (service life) of polymers is generally not so long;

-   Lines 25-26: another cause of plastic pollution problem is polymer deterioration (e.g. deterioration of paints, that produce an high amount of plastics/microplactics in the environment);

-    Line 44: generally, polymers are organic compounds (e.g. PE, PVC …). Authors must specify what kind of “organic alternatives” are considering;

-       Line 49: replace “reuse” with “recycle”;  

-       Sentence in lines 56-58: please rephrase it more clearly;

-       Figure 1: add the number of the reference in the figure caption and move the source in the references at the end of the paper;

-        Lines 72-73: delete the “definition of polymer” not so correct and not necessary;

-    Lines 81-82: the properties of bioplastics are not as similar to those of plastics obtained from oil (e.g. mechanical properties), Authors have to clarify;

-       Line 84: delete “help”;

-       Line 88: delete “helping the European Union to meet its greenhouse gas targets”, this is true all over the world;

-       Line 97: replace “bioplastics” with “plastics”;

-      Lines 97-99: Authors must consider that Bisphenol A (not Biosphenol A) is used only for the production of some (few) plastics;

-        Line 101: replace “industrial application” with “application in construction field”;

-      Figure 2: not organic-based, but Bio-based (all polymers are generally organics); add the number of the reference in the figure caption and move the source in the references at the end of the paper;

-          Line 117: remove “a type of plastic”;

-       Sentence in lines 129-133: it’s not clear if the study reported in [8] excluded or not the “oxo-biodegradable plastics”;

-       Section 2.2.3: please add some examples of this plastics;

-       Line 140: the first sentence is a repetition, already reported, please delete it;

-       Line 145: delete “of derivation”;

-       Line 158: fossil-based plastics is generally obtained from oil extraction (not “mining”);

-       Lines 158-164: Authors have to consider that also wood that is used in construction is a “cultivated material” and this is not the sole example (in the past, a lot of cultivated/”natural” materials were used, nowadays substituted by other materials with improved properties);

-       Line 174: organic recycling is not a “new disposal methods”, it’s used not only for bioplastics;

-       Lines 188-189: delete “in order for it”;

-       Line 190: Author must consider that methane is “organic”, not “inorganic”;

-       Lines 193-195: the “energy recovery” is commonly considered also for fossil-based plastics

-       Lines 198-200: it is suggested to report: although these materials can be effectively recycled, in many countries ... ;

-       Line 202: replace “dry mixes” with “mortar”;

-       Line 210: replace “biological” with “bio-based”; delete “dry”;

-       Line 213: the value reported for the percentage (15% in volume) of admixtures in concrete is very high, Authors should check the reference;

-       Line 217: delete “which results in concrete with good malleability”;

-     Lines 218-219: replace “and a correspondingly high when it comes to compressive strength” with “thus high mechanical properties and durability can be obtained”;

-       Line 224: replace “high” with “also”;

-       Lines 225-226: it’s not clear the effect of the Welan rubber; is this a viscosity modifying agent?

-       Line 227: delete “dry”;

-       Line 242: replace “mortar” with “lime mortar”;

-   Lines 244-246: Authors must consider that the possible use of this materials in concrete have to be evaluated considering the possible negative effects on concrete setting and hardening;

-       Line 251: not “adhesive” but “binders”;

-       Lines 251-252: delete “within the building material”;

-       Line 254: replace “with adhesive behaviour” with “as a binder”;

-       Lines 255-258: add the temperature of heating;

-       Line 258: delete “to the ether”;

-            Line 277: delete “and shown in the research cited earlier” and add [15];

    Line 283: replace “commercialization” with “its use”;

-     Line 288: Authors must consider that expanded clay and perlite are not “natural” but obtained with an industrial process (with high energy consumption);

-       Line 298: delete “mentioned earlier” and add reference;

-       Lines 352-360: move to 3D printing section;

-       Line 366: delete “alloys”;

-       Line 375: replace “in everything” with “for different applications”;

-       Line 391: correct “form-ma”,

-       Line 407: replace “petrochemical” with “fossil-based”;

-       Move sentence in lines 426-431 at the end of the sentence in line 421;

-       Move the example of use in architecture in lines 421-422 at the end of the section, adding the reference;

-       Line 431; correct “m2”;

-       Add the number of the pages in the references: 14, 18, 22, 25 and 26.

English can be improved.

Author Response

Thanks.

Best regards

Reviewer 2 Report

Please, see the attached file "Review MS. Ref. .encyclopedia-1819449.pdf"

Author Response

Thanks.

Best regards

Round 2

Reviewer 1 Report

no other comments

Reviewer 2 Report

The authors have adequately and carefully addressed the reviewer's comments and significantly improved their manuscript. The manuscript is now well presented and complete, well written, and duly illustrated.

Final assessment

The entry manuscript is now suitable for publication.